# Immunocyte Membrane-Coated Nanoparticles for Cancer Immunotherapy

**DOI:** 10.3390/cancers13010077

**Published:** 2020-12-30

**Authors:** Ping Gong, Yifan Wang, Pengfei Zhang, Zhaogang Yang, Weiye Deng, Zhihong Sun, Mingming Yang, Xuefeng Li, Gongcheng Ma, Guanjun Deng, Shiyan Dong, Lintao Cai, Wen Jiang

**Affiliations:** 1Guangdong Key Laboratory of Nanomedicine, Shenzhen Engineering Laboratory of Nanomedicine and Nanoformulations, CAS-HK Joint Lab for Biomaterials, CAS Key Laboratory of Health Informatics, Institute of Biomedicine and Biotechnology, Shenzhen Institute of Advanced Technology, Chinese Academy of Sciences, Shenzhen 518055, China; pf.zhang@siat.ac.cn (P.Z.); zh.sun@siat.ac.cn (Z.S.); gc.ma@siat.ac.cn (G.M.); gj.deng@siat.ac.cn (G.D.); lt.cai@siat.ac.cn (L.C.); 2Department of Radiation Oncology, University of Texas Southwestern Medical Center, 2280 Inwood Road, Dallas, TX 75235, USA; yifan.wang@utsouthwestern.edu (Y.W.); zhaogang.yang@utsouthwestern.edu (Z.Y.); weiye.deng@utsouthwestern.edu (W.D.); mingming.yang@utsouthwestern.edu (M.Y.); xuefeng.li@utsouthwestern.edu (X.L.); shiyan.dong@utsouthwestern.edu (S.D.); 3Yantai Yuhuangding Hospital, Yantai 264000, China

**Keywords:** immunocyte membrane-coated nanoparticles, biomimicry, cancer immunotherapy, macrophage, T-cell, natural killer, dendritic cell

## Abstract

**Simple Summary:**

Cancer immunotherapy is a breakthrough in cancer treatment. Unfortunately, despite the encouraging results in clinical treatment, cancer immunotherapy such as CAR-T, PD-1 still faces lots of challenges. Therefore, it is necessary to develop new methods to improve the effectiveness and safety of tumor immunotherapy. In recent years, cell membrane-coated nanomaterial is one of the most promising drug delivery systems and is receiving a great deal of attention due to its naturally biocompatible characteristics. This review summarizes the latest research progress, the advantages, the disadvantages, and the application of immunocyte membrane-coated nanoparticles in cancer immunotherapy.

**Abstract:**

Despite the advances in surface bioconjugation of synthetic nanoparticles for targeted drug delivery, simple biological functionalization is still insufficient to replicate complex intercellular interactions naturally. Therefore, these foreign nanoparticles are inevitably exposed to the immune system, which results in phagocytosis by the reticuloendothelial system and thus, loss of their biological significance. Immunocyte membranes play a key role in intercellular interactions, and can protect foreign nanomaterials as a natural barrier. Therefore, biomimetic nanotechnology based on cell membranes has developed rapidly in recent years. This paper summarizes the development of immunocyte membrane-coated nanoparticles in the immunotherapy of tumors. We will introduce several immunocyte membrane-coated nanocarriers and review the challenges to their large-scale preparation and application.

## 1. Introduction

Cancer immunotherapy (immuno-oncology) is a kind of treatment that aims to restore the capacity of the immune system to identify and reject cancer. Immunotherapy is considered to be a promising new generation of therapy, since immunotherapy can eliminate cancer cells by activating adaptive immunity and innate immunity of patients, with higher specificity and less toxicity, Compared with traditional therapies, such as chemotherapy and radiotherapy [1,2]. Presently, cancer immunotherapy mainly includes cellular immunotherapy (Provenge, CAR-T), antibody therapy (Alemtuzumab, Durvalumab), cytokine therapy (interferon, interleukin) and oncolytic viruses. However, despite the encouraging results in tumor treatment, cancer immunotherapy still faces lots of challenges, which may be mainly attributed to tumor heterogeneity, immune cell dysfunction, tumor microenvironment, acquired resistance to immunotherapy, and immunotoxicity [3]. Therefore, it is necessary to improve the effectiveness and safety of tumor immunotherapy. Recent trends in cancer immunotherapy have focused on developing immunocyte membrane-based nanomaterials.

Cell membranes are composed primarily of lipids, proteins and carbohydrates, and they give cells their structure, protect intracellular components from the extracellular environment, and regulate the materials that enter and leave the cell [4,5,6,7]. The cell membrane also plays an important role in cell-cell contact, surface recognition, and cell signaling and communication [8,9]. The protein content of cell membrane is very high, usually about 50% of membrane volume [10]. These membrane proteins are crucial to the cellular survival and function because they are responsible for many vital biological events, such as energy storage, cytoskeleton contact, signaling, enzymatic activity, substance transport, and information transduction [11,12]. Moreover, the membrane proteins can differ substantially across different cell types, and even the same type of cells from different individuals can have completely different glycosylation modifications [10,13]. Hence, as membrane markers, membrane proteins and their glycosylation, which allow cells to recognize each another, are of great importance for cell-to-cell communication. On the one hand, cell-cell recognition is critical for cellular signaling processes that can affect formation and development of tissues and organs in early stage of ontogeny. On the other hand, cell-to-cell communication based on membrane proteins plays a very key role in the distinction between “self” and “non-self” in subsequent immune responses [14].

As a lipid bilayer mixed with proteins, the cell membrane is actually a perfect two-dimensional nanomaterial with various functions, since the thickness of cell membrane is only about 10 nm [15]. Moreover, owing to the lipid bilayer’s spontaneously “self-sealing” behavior, broken cell membranes can also naturally form nearly spherical nanovesicles with an internal, aqueous lumen [10]. It is feasible and would be significant to use cell membranes to coat nanomaterials for more effective drug delivery. Numerous nanomaterials coated with cell membrane have been fabricated from many different types of cells, such as red blood cells [16,17,18,19,20], cancer cells, immunocytes, stem cells, platelets [21], and bacteria. These cell membrane-based biomimetic nanomaterials not only retain the complex biological functions of natural cell membranes, but they also maintain the highly adjustable physicochemical properties of the synthesized nanomaterials [22,23,24].

Natural cell membranes that camouflage the nanoparticles’ antigenic diversity from the source cells can have a variety of source cell-relevant functions, such as “self” markers, biological targeting, communication and negotiation with the immune system, and homing to specific regions [25,26,27]. Cell membrane-coated nanoparticles’ unique abilities to biomimic and biointerface with cell membranes not only give them certain physiochemical properties, such as high cargo loading and great stability under high shear-stress conditions [24,28,29], but also make them tunable to have certain biological functions, such as long circulation, targeted recognition, enhanced accumulation in disease sites, and deep tumor penetration [30,31,32].

In addition, bare nanoparticles usually adsorb biomolecules in plasma and/or intracellular fluid in vivo and form a biological coating on their surface, namely protein corona owing to high surface free energy of nanomaterials [33]. The composition of protein corona varies depending on the composition, size, and surface modification of the nanomaterial, as well as the environment surrounding the nanomaterial. This protein corona may shield the specific surface structure and cover the targeting ligand, thus hindering the specific reaction between the nanoparticle and its target, which greatly affects the nanoparticles’ fate and may results in removal of nanoparticles from the bloodstream [34,35,36]. Therefore, it is crucial to avoid the formation of protein corona in the development of nanomaterials for biological or biomedical application. However, having a cell membrane coating around the nanoparticles can successfully prevent nanomaterials from forming a protein corona. As the interface between the cell and the outside world, the cell membrane is perfectly compatible with biofluids, effectively blocking the interactions between the nanomaterials and biological system. This strategy allows nanoparticles to navigate more effectively within the body, thereby limiting off-target side effects, significantly regulating immune responses and, ultimately, enhancing treatment efficacy and expanding the application range of nanomaterials [37,38].

Herein, we summarize the recent progress in research on biomimetic immunocyte membrane-coated nanoparticles for cancer immunotherapy. This article will introduce macrophage membrane, T-cell membrane, Natural killer membrane and dendritic cell membrane-based nanoparticles (neutrophil membrane and platelet membrane-based nanomaterials have been introduced in a recent review, so this article will not discuss these in detail). We will highlight their novelty, analyze their potential prospects in the biomedical field, and finally discuss the challenges in their large-scale preparation and application.

## 2. Immunocyte Membrane Molecules Contributing to Nanomaterials’ Anti-Tumor Immune Effects

Immune response depends on the communication and mutual recognition between immunocytes, and between immunocytes and other cells. Immunocyte recognition is based on immunocyte membrane molecules, usually known as cell surface markers. There is a wide variety of immunocyte surface markers related to tumor immunity, including receptors, antigens, adhesion molecules, and other molecules on the cell surface. Table 1 summarizes some of the major cell membrane surface markers that may contribute to nanomaterials’ anti-tumor immune effects. Because of these specific surface markers, immunocyte membranes have unique functions and can play a special role in assisting drug delivery, especially in tumor cell recognition and anti-tumor immunity.

Because nanoparticles are foreign substances, one of the fundamental problems and technical barriers to using them is uptake by the reticuloendothelial system (RES) or the mononuclear phagocytic system (MPS), which is part of the immune system and consists of phagocytic cells such as monocytes, macrophages in lymph nodes and spleen, and Kupffer cells in liver [39,40,41]. When nanoparticles enter the body, they are first “opsonized” and coated by non-specific proteins that make them more recognizable to phagocytic cells such as macrophages, monocytes, and dendritic cells. Once opsonization, phagocytosis will occur, by which the nanoparticles are engulfed and eventually destroyed or removed from the bloodstream [42]. The most common strategy to reduce RES uptake is to shield nanoparticles with polyethylene glycols (PEGs) or other polymers. This is effective, but it cannot avoid uptake completely [43]. However, when nanoparticles are covered with immunocyte membranes, especially macrophage membranes, the RES system can be completely avoided, because the immune system recognizes nanoparticles camouflaged with macrophage membranes as “self” rather than “foreign.”

In addition, some specific receptors and adhesion molecules on macrophage membranes, such as C-C chemokine receptor 2 (CCR2), vascular cell adhesion molecule-1 (VCAM-1, CD106), and intercellular adhesion molecule-1 (ICAM-1; CD54), can guide nanoparticles coated with membranes to inflammatory and tumor sites. Notably, the binding of ICAM-1with macrophage adhesion ligand-1 (Mac-1; ITGAM), leukocyte function-associated antigen-1 (LFA-1), and fibrinogen can facilitate the transmigration of leukocytes across vascular endothelia [44,45]. ICAM-1 and soluble ICAM-1 also have antagonistic effects on the tight junctions and thus promote nanoparticles coated with membranes to cross the blood–brain barrier (BBB) [46,47].

The second valuable cell membrane is natural killer (NK) cell membranes. NK cells are unique lymphocytes that can recognize and kill aberrant cells without antibodies or major histocompatibility complex (MHC) [48]. Therefore, NK cells are key for tumor cell surveillance, because tumor cells that are missing MHC I markers cannot be detected and destroyed by other immune cells, such as T-cells [49]. Although NK cells lack antigen-specific cell surface receptors, they have many alternative receptors that can recognize tumor cells, including NKG2D, NKp44, NKp46, NKp30, and DNAM [50,51]. NKG2D, for example, is a disulfide-linked homodimer that recognizes several ligands, including UL16-binding protein (ULBP) and MHC class I chain-related gene A (MICA), which are typically expressed on tumor cells [52].

T-cell membranes are helpful for recognizing and targeting tumor cells because of the T-cell receptor (TCR) on the cell surface. As a protein complex, TCR recognizes fragments of antigens as peptides bound to MHC molecules, thus allowing T-cells to target both surface and intracellular tumor neoantigens [53,54]. Neoantigens are mutated antigens specifically expressed by tumor tissue, but not expressed on the surface of normal cells, so they are highly specific for individuals [55]. Since TCRs can recognize neoantigens and then target tumor cells or tumor tissue, nanoparticles coated with T-cell membranes could target tumor cells or tumor tissue with highly specificity for individuals.

Furthermore, PD-1, CTLA4, and other specific checkpoint inhibitory receptors on T-cells are harmful to tumor immunotherapy, but these checkpoint inhibitory receptors can also identify the corresponding ligands, such as PDL-1, on tumor cells [56,57]. Hence, TCRs are beneficial to the delivery of nanodrugs, because they can target tumor cells.

The greatest advantage of mature dendritic cell (DC) membranes is that they possess the antigen presentation function of whole DCs and can specifically activate T-cells, because they have a broad spectrum of peptide/MHC complexes on their surface [58,59]. DCs can also express co-stimulatory molecules and adhesion molecules, such as ICAM-3/CD50, CD40, CD44, and integrin family members [60,61]. These markers can reduce the negative charge on the cell surface and mediate cell adhesion, thus promoting the interaction between DCs and T-cells [62].

## 3. Macrophage Membranes

Macrophages express a wide variety of surface membrane receptors that recognize a wide range of endogenous and exogenous ligands [63]. It is through the mediation of these membrane receptors that macrophages can interact not only with natural and varied self-components of the host, but also with foreign components, such as microbes, and then induce appropriate responses [64,65]. Hence, nanoparticles coated with macrophage membranes is really useful for innate and acquired immunity.

### 3.1. Immune Evasion and Tumor Targeting

Su et al. [66] have developed Saikosaponin D loaded macrophage membrane-biomimetic nanoparticles (SCMNPs) for breast cancer therapy. This nano-drug system consists of a poly (latic-co-glycolic acid) nanoparticles and macrophage membrane hybridized with T7 peptide. The presence of receptors and ligands on macrophage membranes endowed SCMNPs with the ability to macrophage-homing, while the presence of T7 peptide enabled the nanoparticles to recognize tumor cells overexpressing transferrin receptors. SCMNPs thus could escape phagocytosis by the RES system and selectively accumulate into tumor tissues. The drug Saikosaponin D encapsulated in SCMNP could inhibit tumor neovascularization by regulating angiogenic pathway related factors such as vascular endothelial growth factor (VEGF), Phosphatidylinositol 3-kinase (PI3K)/protein kinase B (AKT), and extracellular-regulated kinase (ERK), thus effectively inhibiting tumor growth and metastasis of breast cancer. This biomimetic strategy based on macrophage membranes provided a target anti-angiogenic therapeutic model for the precise and effective treatment of breast cancer.

In addition, Yan et al. [67] developed macrophage membrane-cloaked luminescence nanoparticle@MOF-derived mesoporous carbon nano-drug delivery system to escape the RES system. This multiple drug co-loaded nano-drug delivery system exhibited potential for autofluorescence-free, long-lasting persistent fluorescence imaging-guided drug targeted delivery and cancer therapy. Li et al. [68] subsequently reported that liposomes coated with isolated macrophage membranes could enhance delivery to metastatic sites via α4 integrin−VCAM-1 interactions and then target lung metastasis of breast cancer. Jiang et al. [69] collected macrophage membranes to encapsulate cskc-PPiP/PTX@Ma nanoparticles and found that this delivery system exhibited a favorable tumor-homing ability in systemic circulation and high biocompatibility because of its membrane coating.

Recently, Liu et al. [70] used macrophage membrane-coated iron oxide nanoparticles (Fe_3_O_4_@MM) to enhance photothermal tumor therapy. Because they were derived from an Fe_3_O_4_ core and had a macrophage membrane shell, the Fe_3_O_4_@MM NPs exhibited good biocompatibility, immune evasion, cancer targeting, and light-to-heat conversion capabilities. Quercetin-loaded hollow bismuth selenide nanoparticles are another example of nanoparticles coated with macrophage membranes. Compared with bare nanoparticles, macrophage membrane-camouflaged nanoparticles prolonged circulation life and accelerated and enhanced tumor tropic accumulation, thus showing promise in suppressing breast cancer lung metastasis in vivo [71]. Recently, Deng et al. [72] constructed a multifunctional biomimetic superparticle, termed as DOX-QDs-Lip@M nanoparticle for enhanced cancer imaging and anti-metastasis treatment. Due to the presence of α4 integrins in macrophage membrane, the nanocarrier had the ability to bind VCAM-1 on cancer cell and escape the immune system’s response. Furthermore, the bionic membranes could stabilize the synthetic liposome structure and thus avoid the leakage of the loaded content such as DOX and ZAISe/ZnS QDs in the liposome.

Mesoporous silica nanocapsules camouflaged by macrophage cell membranes actively targeted tumors because of the guidance of the surface proteins on the macrophage cell membranes [73]. Because coating with macrophage cell membranes is a simple and effective surface engineering method that can activate tumor targeting, these membranes have also been used to disguise conversion nanoparticles (UCNPs) [74]. Xia et al. [75] developed albumin nanoparticles coated with macrophage plasma membranes loaded with paclitaxel to prolong blood circulation and achieve targeted therapy against malignant melanoma.

### 3.2. Penetrating the Blood–Brain Barrier (BBB) and Targeting Glioblastoma

BBB is a crucial selective semi-permeable border, which composed of endothelial cells adjoined continuously by the tight junctions and can restricts the passage of solutes. However, BBB also limits beneficial drug delivery to the central nervous system. Thus, overcoming the obstacle of BBB is the primary challenge in the treatment for central nervous system disorders. We developed IR-792 nanoparticles (MDINPs) decorated with macrophage plasma membranes for NIR-Ib fluorescence imaging-guided photothermal therapy (PTT) on orthotopic glioblastoma [76]. The macrophage membrane proteins (Integrin α4 and Mac-1) in the outer layer of MDINPs bind to the corresponding receptors, such as VCAM-1 and ICAM-1, on brain vascular endothelial cells, which activate the signaling pathway, reduce the expression of tight junction-related proteins (zonula occludens 1), and finally break down the tight junction, so that MDINPs can easily pass through the BBB. MDINPs also selectively accumulated at the tumor site because of the tumor targeting effect of the macrophage membranes. In animal experiments, MDINPs-mediated NIR-Ib fluorescence imaging-guided PTT prolonged the survival of mice. These results show a new strategy for integrating diagnosis and therapeutics in glioblastoma.

### 3.3. Anti-Proliferation

As a precursor of the soluble form of TNFα, transmembrane TNFα is expressed on the membrane of some cells such as lymphocytes and activated macrophages [77]. Since the 1980s, studies have shown that transmembrane TNFα expressed on human macrophages and lymphocytes induces strong and long-term tumor regression [78,79,80,81]. A good example is that tumor cells could be lysed by incubating them with transmembrane TNFα on paraformaldehyde-fixed activated macrophages [80]. Based on macrophages’ ability to easily produce TNFα by the induction with lipopolysaccharide or other agents, a unique therapeutic nano-drug delivery system had been prepared by cloaking a degradable, biosafety, chitosan nanocarrier with bioengineered TNFα-binding macrophage membrane [82]. In this paper, THP-1 cells (a human monocytic cell line derived from an acute monocytic leukemia patient) differentiated with phorbol 12-myristate 13-acetate (PMA) were induced by bacterial lipopolysaccharide to produce macrophage membrane-tethered TNFα first. Then, the TNFα-binding macrophage membrane were decorated onto the surface of polymeric nanoparticles. In vitro experiments have shown that the membrane-cloaked nanocarriers have high stability and biocompatibility, and have the potential to significantly prevent the proliferation of tumor cells.

### 3.4. Macrophage Hybrid Membrane

Although macrophage-based tumor immunotherapy approaches have many advantages, they also face some challenges. First, tumor cells in general express CD47 molecule, an inhibitory receptor, which interacts with the signal-regulated protein alpha (SIRPα) on macrophages to prevent phagocytosis of tumor cells by macrophages. Second, an immunosuppressive microenvironment in tumors polarize macrophages from the anti-tumor M1 phenotype to the tumorigenic M2 phenotype. M2 macrophages can promote tumorigenesis, vascular regeneration, and metastasis. However, a hybrid cellular membrane nanovesicle (hNVs) reported by Chen et al. could overcome these drawbacks of macrophage-based immunotherapy and amplify macrophage immune responses against tumor recurrence and metastasis (Figure 1) [83]. hNVs, originated from three types of membranes, macrophage, platelet, and cancer cell, could interact with circulating tumor cells, accumulate in surgical sites, and then repolarize tumor-associated macrophages towards M1 phenotype. At the same time, hNVs also could bind to macrophages via CD47 and block CD47–SIRPα interaction between tumor cells and macrophages. As a result, hNVs could promote macrophages phagocytosis of tumor cells and significantly enhance the efficiency of cancer immunotherapy.

Yuan et al. [84] also explored macrophage-cancer hybrid membrane-coated nanoparticles for targeted treatment of lung metastases from breast cancer. Owing to the presence of macrophage and cancer cell membranes, these nanodrugs had a multi-targeting capability, and consequently they could accumulate to sites of inflammation, as well as target homogenous metastatic tumor. The doxorubicin-loaded nanodrugs were highly effective in the treatment of cancer, with nearly 88.9% cure rate in breast cancer-derived lung metastasis model.

## 4. T-Cell Membranes

T lymphocytes play a central role in the immune response. T-cells can attack and destroy tumor cells by their highly specific TCR, which bind to antigens present on the surface of other cells [85,86,87,88,89]. Each T-cell recognizes only a single antigen, but collectively, T-cells have a wide array of receptors targeting millions of antigens [90]. Many immunotherapy approaches have been developed to get rid of cancer cells by activating TCR-peptide-MHC interactions. Adoptive T-cell therapies such as chimeric antigen receptor (CAR) T-cell therapy have been a huge clinical success in hematological cancer treatment. However, such methods are generally not clinically effective in solid tumor treatment because of the lack of tumor-specific biomarkers on the surface of solid tumor cells [91].

### 4.1. Targeting Tumors through TCRs

Recently, Nguyen et al. [92] explored a new approach: coating Trametinib-loaded PLGA nanoparticles with melanoma-specific T-cell membranes (T-MNPs) to improve the therapeutic efficiency of chemo-drugs and overcome the non-specific targeting of anticancer drugs. The T-cell membranes on the surface of T-MNPs were derived from the T-cell hybridoma, which could express gp100 antigen-MHC molecule, could specifically bind melanoma cells expressing gp100 peptide, and enhance the uptake of T-MNPs. The T-cell membrane also gave the nanoparticles high stability and hemo-compatibility and cyto-compatibility. Membrane-coated NPs were taken up by a melanoma cell line in vitro three times as much as bare nanoparticles. Moreover, in vivo biodistribution studies displayed the theragnostic capabilities of these NPs, the tumor retention of which was more than twice as high as the uncoated and non-specific membrane-coated groups. Hepatocellular carcinoma (HCC)—specific CAR-T cell membrane-coated nanoparticles have been developed by Yuan et al. [93] for the treatment of HCC. The nanosystem packed with CAR-T cell membranes showed a superior ability to target HCC cells and improved therapeutic effect compared to naked nanosystem, because the CAR-T cell membranes specifically recognize GPC3+ HCC cells.

### 4.2. The Dual-Targeting Strategy

Since T-cell membranes have a variety of specific recognition receptors, these membranes are ideally suited for biomimetic nano-drug systems to tumors. This single-targeted therapy strategy is not so effective because of tumors’ inter- and intra-heterogeneity. Hence, dual or multi-targeting is a promising approach to tackle either target antigen loss or down-regulation [94]. Cai et al. [95] employed a dual-targeting strategy based on azide (N_3_)-labeled T-cell membrane-coated nanoparticles to enhance PTT for tumor. In this work, the bicyclononyne (BCN) group as artificial receptors were introduced into the tumor via natural glycometabolic labeling by pretreating the tumor with Ac_4_ManN-BCN group first. Then, T-cell membranes were modified with N_3_. Therefore, the nanoparticles coated with N_3_-labeled T-cell membranes (N_3_-TINPs) could dual-target tumor cells via both N_3_-BCN bio-orthogonal click reaction and TCR-peptide/MHC recognition. The fluorescence intensity of the mouse tumors treated with N_3_-TINPs was 1.5-fold that of unlabeled nanoparticles. The accumulation of N_3_-TINPs in the tumor obviously increased the photothermal curative efficacy, yet virtually no side effects. Thus, dual-targeting N_3_-TINPs-based click chemistry approach could offer an alternative dual-targeting strategy for advancing cancer treatment.

## 5. NK Cell Membranes

NK cells can lead to immune surveillance against cancer and eliminate small tumors early. Because they can engage tumor targets without needing specific antigens, NK cells’ therapeutic potential has been broadly explored to control the metastatic dissemination of malignancies [96]. NK cells lack genetically rearranged antigen receptors [97], but they can still recognize and directly lyse abnormal cells without prior sensitization [98,99]. This occurs through a well-established and unique set of receptors expressed on NK cells’ surface, which trigger cell lysis activity by interacting with ligands on infected and transformed cells [100]. The receptors on NK membranes, especially activated receptors such as NKp30, NKp44 NKp46, DNAM-1 (CD226), and NKG2D, are closely associated with tumor recognition and tumor killing [101]. NKp30 recognizes B7-H6 tumor antigens, while NKp44 binds proliferating cell nuclear antigen (PCNA) and other tumor-associated ligands [100,102]. NKp46 inhibits metastatic growth in mice [103]. DNAM-1 is an adhesion and co-stimulatory molecule that promotes NK cell cytotoxic activity upon binding to its ligands, CD112 and CD155 [104]. The NKG2D form transduces activating signals to initiate cytotoxic activity upon binding to specific stress-induced ligands, MICA and MHC class I polypeptide-related sequence B (MICB), which are selectively expressed on tumor cells [105,106,107].

### 5.1. Targeting Tumors

Aryal Murali et al. developed an NK cell membrane-cloaked fusogenic liposomal delivery system (NKsome) for targeted cancer therapy [108]. The engineered NKsomes have a variety of receptor proteins on their surface, thus they exhibited not only high binding ability to tumor cells in vitro, but also powerful tumor-homing efficiency in vivo. They also have outstanding biocompatibility and long blood circulation time (18 h). Further in vivo experiments showed that NKsome had promising therapeutic effects on MCF-7 tumor xenograft models. This study demonstrated that biomimetic nanocarriers based on NK cell membrane can partly communicate like immune cells and thus have therapeutic advantages through enhancing tumor drug delivery.

### 5.2. M1 Polarization and Induction of Immunogenic Cell Death (ICD)

Our group reported on NK cell membrane-cloaked photosensitizer TCPP-loaded nanoparticles (NK-NPs) that could eliminate primary tumors and inhibit distant tumors [109]. NK cell membranes on the surface of NPs enabled the NK-NPs to target 4T1 tumors via NK cell membrane receptors, such as NKG2D and DNAM-1 (Figure 2). Furthermore, NK cell membrane proteins such as RAB-10, IRGM1, RANKL Galectin-12, and CB1, can interact with macrophage surface receptors such as tumor necrosis factor or receptor toll-like receptor 4, to prompt or increase pro-inflammatory M1 macrophage polarization, which would kill tumor cells directly by secreting reactive oxygen species (ROS) and nitrogen radicals [110,111,112].

At the same time, the photosensitizer TCPP inside the NPs, after being irradiated with light, can destroy the cancer cells and induces immunogenic cell death (ICD). These dying cancer cells secrete damage-associated molecular patterns (DAMPs), promote activation of DCs and initiate an adaptive immune response. Results from animal experiments confirmed that NK-NPs selectively accumulate in 4T1 tumors and could eliminate primary tumor growth and produce an abscopal effect (a distant anti-tumor activity induced by local treatments), and then inhibit distant tumors. Thus, this NK cell membrane-based method offers a promising strategy for tumor immunotherapy.

### 5.3. Penetrating the BBB

Treating brain tumors with drugs is extremely difficult because the BBB hinders the delivery of systemic therapies into the tumor. Many strategies have been proposed for improving drug delivery across the BBB [113,114]. Since immunocytes, such as macrophages, neutrophils, T-cells, and NK cells, use specialized mechanisms to penetrate the BBB without compromising their structural integrity [115,116], using these immune cells as vehicles to deliver drugs through the BBB has recently become a research hotspot in this field.

Our group developed aggregation-induced emission (AIE) characteristic nanodots decorated with NK membranes (NK@AIEdots) for near-infrared-II fluorescence-guided glioma theranostics [117]. The binding of NK@AIEdots with cell adhesion molecules (CAMs) on brain microvascular endothelial cells by LFA-1 (lymphocyte function-associated antigen 1) and VLA-4 (very late antigen-4) on NK cell membranes could trigger an intracellular signaling cascade, which would disrupt tight junctions and reorganize actin cytoskeletons to form intercellular gaps at the BBB. Then, NK@AIEdots could cross the BBB through the paracellular pathway. Furthermore, NK@AIEdots could target malignant glioma cells (U-87 MG) through the receptors (DNAM-1 and NKG2D) on the NK membrane. The sufficient accumulation and high quantum yield of NK@AIEdots, on the one hand, allow them to perform high contrast and through-skull tumor NIR-II fluorescent imaging, and on the other hand, create NK@AIEdots-induced localized hyperthermia effects with laser irradiation, which could effectively inhibit glioma growth.

## 6. Dendritic Cell Membrane

As the most potent of all professional antigen-presenting cells (APCs), dendritic cells (DCs) can activate not only resting helper T-cells, but also memory and naive T-cells [60,61,118,119,120]. DCs act as immune sentinels that survey the body and collect information for responding to challenges, and they play a central role in initiating and regulating adaptive immune responses [121]. The generation of tolerogenic DCs induces anergy and contributes to tumor cells escaping immune surveillance, so it would be significant to artificially manipulate DCs to promote T-cells [122]. There are several approaches used in clinical trials to generate DCs, for example, in vivo expansion of circulating DCs, differentiation from CD34^+^ hematopoietic precursors or monocyte precursors, and more recently, isolation and enrichment of circulating blood DC subpopulations. However, the clinical efficacy of most of these methods remains unsubstantiated. There is great potential for the development of a new DC-based cancer immunotherapy. At present, a DC membrane-based therapeutic method has been developed.

### 6.1. Activation and Maintenance of Antigen-Specific T-cells

Moon et al. [123] developed nanosized dendritic cell membrane vesicles (DC-MVs) capable of activating APCs and delivering peptide antigens. DC-MVs successfully led to maturation of dendritic cells and promoted T-cell survival and proliferation in vitro. These effects were also observed in vivo, where antigen-specific T-cells adoptively transferred into mice exhibited greater proliferation after vaccination with DC-MVs and peptide antigen than after peptide antigen alone. Additionally, vaccination with DC-MVs enhanced levels of endogenous T-cell responses against model antigen ovalbumin in an OVA-expressing tumor model. These results suggest that DC-MVs are a potentially attractive platform for further development as a peptide-based vaccine for cancer immunotherapy.

Zhang et al. [124] reported on DC-cancer fused membrane-based nanoparticles (NP@FMs) for targeted tumor therapy. The fused membranes (FMs) not only endowed NP@FMs with targeting capability to homologous tumors (breast cancer cell line 4T1), but also provided them with the capability to locate the lymph node and induce immune response. NP@FMs could trigger innate immunity and initiate adaptive immunity, they also could induce death of cancer cells by photodynamic therapy (PDT). Owing to the combination of PDT and immunotherapy, NP@FMs displayed powerful ability to inhibit the proliferation of distant tumors without radiation exposure. Furthermore, the primary and distant tumors were almost completely eliminated.

### 6.2. Intelligent Nano-DCs

Our group developed intelligent nano-DCs (iDCs), which consist of nanoparticles loaded with photothermal agents (IR-797) and coated with a mature DC membrane. The DC membrane on the iDCs maintains the antigen presentation and T-cell priming capabilities of native DCs (Figure 3). The iDCs can enter the lymph node and stimulate T-cells, which migrate to the tumor site. The activated T-cells reduce the expression of heat shock proteins (HSPs) in tumor cells, thereby rendering them more sensitive to heat stress. Adding mild PTT (42–45 °C) can enhance the tumoricidal effect. Consequently, dying tumor cells and surviving immune cells can induce ICD, reinitiate the self-sustaining cycle of cancer immunity, and contribute to a synergistic anti-tumor effect. Furthermore, unlike the adoptive transfer of activated DCs, iDCs as a refined and precise system in combination with DC-based immunotherapy and thermal therapy can be stored long-term and at a large scale, so they can be applied in different patients [125].

## 7. Conclusions

We have summarized the current research on immunocyte membrane-coated nanoparticles (Table 2). As a new biomimetic drug delivery platform, the immunocyte membrane covering approach is a feasible way to overcome the limitations of introducing foreign materials into the immune system by providing a unique biological interface, and would thus have a wide range of advantages for activating the innate and adaptive immune responses. The cell membranes discussed herein include those originating from macrophages, NK cells, T-cells, and dendritic cells. Macrophages are the main component cell of the RES, so macrophage membranes can help nanoparticles escape the phagocytosis of RES perfectly. In addition, some specific receptors and adhesion molecules can guide nanoparticles camouflaged with macrophage membranes to inflammatory and tumor sites. NK membranes can guide particles to target tumor sites and cross the BBB by a well-established and unique set of receptors expressed on the surface of NK cells. T-cell membranes display some distinctive properties, including binding to neoantigens and displaying specific receptors, which endow biomimetic nanoparticles with immune escape and good targeting abilities. Modified DC membranes enable the membrane-based nanocarriers to present antigens and then activate T-cells. Moreover, by combining with other therapeutic agents such as chemotherapeutic drugs or photosensitizers to treat tumors, cell membrane-coating approaches could enhance cancer therapy.

However, there are some critical issues that must be addressed for the further development and translation of immunocyte membrane-based nanocarriers. First, there is an urgent need for a standard protocol to guide the preparation of immunocyte membrane-based carriers, owing to the difficulty of controlling the parameters of natural materials. Since the functional proteins on cell membranes are susceptible to inactivation, the extraction of cell membranes needs to be done carefully. The whole membrane extraction mainly includes cell lysis, removal of intracellular contents, purification of membranes, and coating cell membrane onto nanoparticle core. The detailed preparation parameters vary according to the cell type. To generate cell membranes-derived nanoparticles with reproducibility and scalability, a more standard protocol needs to be proposed to guide the synthesis of biomimetic nanomaterials.

Second, more researchers should study the integrity of membranes on artificial particles in vivo. The incomplete membrane may lead to the exposure of naked nanomaterials, which in turn may affect the biological effectiveness of cell membranes-derived nanoparticles. There are no relevant research data concerning the extent to which the non-integrity of cell membrane actually affects the function of cell membranes-derived nanomaterials. Therefore, more research should be done in the integrity of membranes on nanoparticles.

Third, measures should be taken to further transform and functionalize immunocyte membranes to broaden the application of membrane-derived nanoparticles. These methods, such as membrane hybridization, lipid insertion, metabolic engineering, and genetic modification, would contribute diverse functions in a nondisruptive fashion while preserving the natural function of the cell membranes. In addition, Immunocyte-derived exosomes also are considered to be very promising system for drug delivery and are receiving a great deal of attention due to their naturally biocompatible characteristics [126]. The exosomes based on monocytes and macrophages could prolong blood circulation time and avoid immune phagocytosis [127]. DC-derived exosomes show attractive application prospects in vaccine delivery, and they have been proven safe in multiple phase I trials in different types of cancers [128]. Nevertheless, the difficulty in purifying exosomes maybe restrict their large-scale application.

Finally, the biological safety of immunocyte membranes in nanoparticles in vivo should be investigated carefully. First, the accumulation of nanomaterials with a large proportion in normal tissues can be harmful. Second, mismatch of allogeneic MHC between donor and recipient of immunocyte membranes may lead to serious safety problems. Thirdly, modified membranes may also raise health risks, such as TNF α-binding macrophage membranes, which maybe induce hyperinflammation.

## Figures and Tables

**Figure 1 cancers-13-00077-f001:**
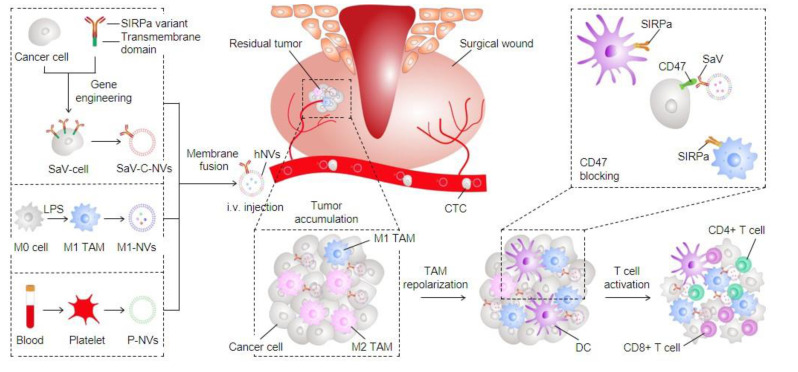
Schematic showing the formation of hNVs and the mechanism by which hNVs amplify macrophage immune responses against cancer recurrence and metastasis [83].

**Figure 2 cancers-13-00077-f002:**
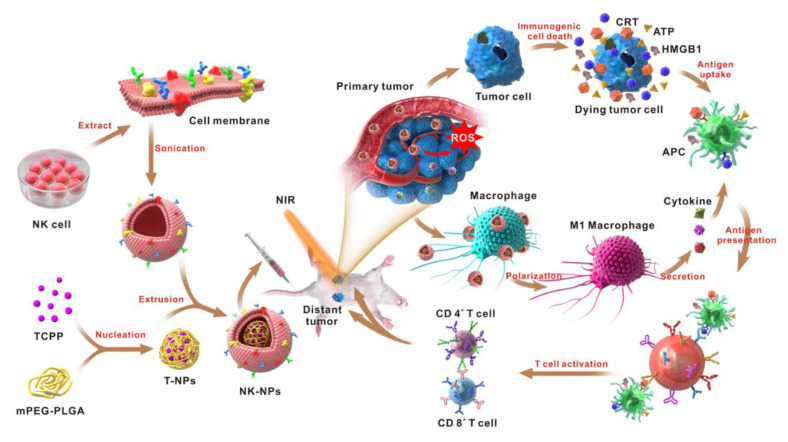
Schematic illustration of NK cell membrane-cloaked nanoparticles for PDT-enhanced cell membrane immunotherapy [109].

**Figure 3 cancers-13-00077-f003:**
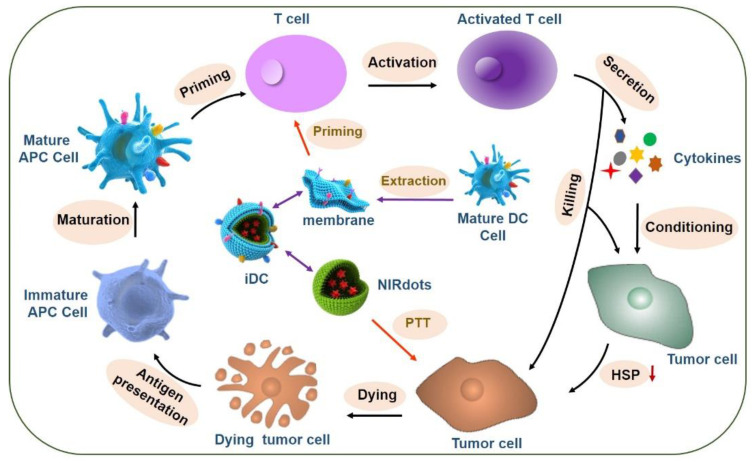
Schematic illustration of the preparation of intelligent dendritic cells (iDCs) and the mechanism of synergy between iDCs and mild photothermal-immunotherapy.

**Table 1 cancers-13-00077-t001:** Immunocyte Membrane Surface Markers That May Contribute to Nanomaterials’ Anti-Tumor Immune Effects.

Cell Type	Marker	Ligand	Function
Macrophage	CCR2	CCL2	Induces a strong chemotactic response, guides immune cells to inflammatory and tumor sites
	VCAM-1	4VLA-4) or integrin α4β1	Cell adhesion, cell signal transduction
	ICAM-1	LFA-1, Mac-1	Facilitates transmigration of leukocytes across vascular endothelia, intercellular adhesion
T-cell	TCR	peptide/MHC complex	Antigen recognition and presentation
	CD28	CD80, CD86	Brings T-cell and antigen-presenting cell membranes into close proximity
	CTLA-4	CD80, CD86	Immune checkpoint and down-regulates immune responses
	PD-1	PD-l, B7	Immune checkpoint and down-regulates immune responses
	LFA-1	ICAM	Cell adhesion and co-stimulator
	LFA-2	LFA-3, CD48	Cell adhesion and co-stimulator
NK cell	NK p46	CD247, FCER1G.	Activates NK cells, mediates tumor cell lysis
	NKp44	NKp44L, 21spe-MLL5, PCNA, HSPGs	Activates NK cells, mediates tumor cell lysis. Transmembrane Signaling Receptor Activity
	NCAM1	rabies virus glycoprotein	MAPK cascade, cell adhesion, host-virus interaction
	FCGR3	immunoglobulin gamma Fc region	Binds to the Fc portion of igg antibodies and activates antibody-dependent cell mediated cytotoxicity (ADCC)
	DNAM-1	PVR, NECTIN2	Signal transducing adhesion involved in the adhesion of certain tumor cells to CTL and NK cells, mediates their cytotoxicity
DC	peptide/MHC complex	TCR	Antigen recognition and presentation
	INAM	IRF3	Stimulates NK cell activation
	ICAM	LFA-1	Cell adhesion and co-stimulator

**Table 2 cancers-13-00077-t002:** The functions, advantages and disadvantages of different immunocyte membrane-coated nanoparticles.

Source of Cell Membranes	Functions	Advantages	Disadvantages	References
Macrophage	Prolonged circulation time; penetrating the blood–brain barrier (BBB); anti-proliferation; tumor targeting; inflammation targeting	Immune evasion; targeting glioblastoma; enhanced intratumoral penetration	Only targeting to limited types of tumor	[66,67,68,69,70,71,72,73,74,75,76,77,78,79,80,81,82]
T-cell	Prolonged circulation time; targeting to specific tumors through TCRs;	Dual-targeting; improved tumoritropic accumulation of drug	MHC Restriction; only targeting to limited types of tumor	[92,95]
NK cell	Prolonged circulation time; tumor targeting; penetrating the blood–brain barrier (BBB); M1 polarization and induction of immunogenic cell death (ICD)	targeting glioblastoma; broad spectrum tumor targeting	Limited multiplication of primary NK cells	[108,109,117]
DC	Antigen-presenting; tumor vaccine; promote T-cells; lymph node targeting	Activation and maintenance of antigen-specific T-cells; providing immunological co-stimulatory molecules	MHC restriction;	[123,124,125]

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
