# Peer review of "Immunocyte Membrane-Coated Nanoparticles for Cancer Immunotherapy"

_cancers, 2020, doi:10.3390/cancers13010077_

Round 1
Reviewer 1 Report
Authors have described the role of membrane coating on nanoparticles for efficient drug delivery purposes. The review is well written and properly formatted to maintain the flow of the key concepts.
Authors also focused on the main difficulties associated with phagocytosis of nanoparticles by immune systems along with build-up of corona proteins, that showed the emerging problem for the use of nanoparticles as drug delivery carriers. The use of different membrane coating technology seems promising for the future design of particles and impose a great application.
I recommend accepting the manuscript after addressing the below comments:
1. In the introduction, authors mentioned the corona protein build-ups in brief, I recommend inclusion of separate paragraph for the description of the topic, mechanism of the build up and main problems associated with the mechanism.
2. Adding a table to describe the main advantages and disadvantage of each method will be beneficial to envision a big picture.
3. Authors should summarize the significance of this topic in details at the end of the review to show why these methods can impose a solution to the real-world problems with nanoparticles, and what can be the main problems associated with the commercialization of the technologies.
Author Response
Reviewer 1:
- In the introduction, authors mentioned the corona protein build-ups in brief, I recommend inclusion of separate paragraph for the description of the topic, mechanism of the build up and main problems associated with the mechanism.
Reply: Thanks for the reviewer’s suggestion. We have made corresponding corrections in the manuscript (Page 2, Line 84-96). Please see the revised version.
- Adding a table to describe the main advantages and disadvantage of each method will be beneficial to envision a big picture.
Reply: Thanks for the reviewer’s suggestion. We have made corresponding corrections and add a table in the manuscript (Page 5). Please see the revised version.
- Authors should summarize the significance of this topic in details at the end of the review to show why these methods can impose a solution to the real-world problems with nanoparticles, and what can be the main problems associated with the commercialization of the technologies.
Reply: Thank you for pointing these out. We have made corresponding corrections in the manuscript (Page 13). Please see the revised version.

Reviewer 2 Report
I think, that presented topic is very interesting and for your journal and readers. The text is written legibly and accompanied by illustrative and nice illustrations. A possible weakness of this approach is reduction in description of discussed studies in the manuscript and thereby an overall overview obtained by reading it.
Therefore I recommend incorporation of other table focused on the biological studies of Immunocyte Membrane–Coated Nanoparticles in the manuscript.
Also subchapter described basic principle of cancer immunotherapy should be incorporated into manuscript.
Author Response
- I recommend incorporation of other table focused on the biological studies of Immunocyte Membrane–Coated Nanoparticles in the manuscript.
Reply: Thanks for the reviewer’s suggestion. We have made corresponding corrections and add a table in the manuscript in the manuscript (Page 5). Please see the revised version.
- Also subchapter described basic principle of cancer immunotherapy should be incorporated into manuscript.
Reply: Thanks for the reviewer’s suggestion. We have made corresponding corrections in the manuscript (Page 1, Line 38-50). Please see the revised version.

Reviewer 3 Report
In the manuscript titled: “Immunocyte Membrane–Coated Nanoparticles for Cancer Immunotherapy” by Gong P. et al. the authors discuss the role of the immunocyte membrane in delivering drugs in oncology.
This is a very interesting and “hot” topic and the review is timely and well written. The figures are well designed and very descriptive.
Additional suggestions:
1) Discuss the potential of artificial exosomes as drug delivery mechanisms.
2) Discuss the role of the spleen as a metabolization hub of synthetic nanoparticles and other targeted therapies.
3) Add a second table in which you summarize the best strategies of synthesizing novel immunocyte membranes.
4) 3.2. please shortly discuss the proprieties of the BBB and how can a drug/nanoparticle cross it.
5) Discuss the risks of using a TNFα-binding macrophage membrane. I am expecting that such a drug could induce hyperinflammation.
6) Page numbering is not correct, please double check.
7) Figure 3 – increase the size of the cells and of the writing.
8) Please expand the last paragraph of the paper, I would suggest to write a full paragraph for each of the 4 points you are discussing: standard protocol, integrity of the membrane in vivo, functionalization of immunocyte membranes and toxicity.
Author Response
Reviewer 3:
- Discuss the potential of artificial exosomes as drug delivery mechanisms.
Reply: Thanks for the reviewer’s suggestion. We have made corresponding corrections in the manuscript. In the discussion part, these contents are supplemented. Please see the revised version.
- Discuss the role of the spleen as a metabolization hub of synthetic nanoparticles and other targeted therapies.
Reply: Thanks for the reviewer’s suggestion. We have made corresponding corrections in the manuscript (Page 3, Line 117). Please see the revised version.
- Add a second table in which you summarize the best strategies of synthesizing novel immunocyte membranes.
Reply: Thank you for pointing these out. We have made corresponding corrections in the manuscript (Page 5). Please see the revised version.
- 2 please shortly discuss the proprieties of the BBB and how can a drug/nanoparticle cross it.
Reply: Thank you for pointing these out. We have made corresponding corrections in the manuscript (Page 7, Line 223-226). Please see the revised version.
- Discuss the risks of using a TNFα-binding macrophage membrane. I am expecting that such a drug could induce hyperinflammation.
Reply: Thank you for pointing these out. In the discussion part, these contents are supplemented (Page13, Line 478). Please see the revised version.
- Page numbering is not correct, please double check.
Reply: Thanks for the reviewer’s suggestion. We have correct the page number in the manuscript. Please see the revised version.
- Figure 3 – increase the size of the cells and of the writing.
Reply: Thanks for the reviewer’s suggestion. We have made corresponding corrections and increase the size of the cells and the writing in the manuscript. Please see the revised version.
- Please expand the last paragraph of the paper, I would suggest to write a full paragraph for each of the 4 points you are discussing: standard protocol, integrity of the membrane in vivo, functionalization of immunocyte membranes and toxicity.
Reply: Thanks for the reviewer’s suggestion. We have made corresponding corrections in the manuscript (Page 7, Line 449-479). Please see the revised version.
